# Making Agriculture Carbon Neutral Amid a Changing Climate: The Case of South-Western Australia

Ross Kingwell 

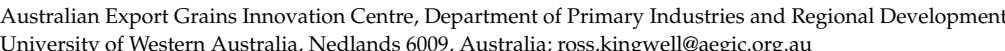

Australian Export Grains Innovation Centre, Department of Primary Industries and Regional Development, University of Western Australia, Nedlands 6009, Australia; ross.kingwell@aegic.org.au

**Abstract:** Making Australian agriculture carbon neutral by 2050 is a goal espoused by several agricultural organisations in Australia. How costly might it be to attain that goal, especially when adverse climate change projections apply to agriculture in southern Australia? This study uses scenario analysis to examine agricultural emissions and their abatement via reforestation in south-western Australia under projected climate change. Most scenarios include the likelihood of agricultural emissions being reduced in the coming decades. However, the impact of projected adverse climate change on tree growth and tree survival means that the cost of achieving agricultural carbon neutrality via reforestation is forecast to increase in south-western Australia. Agricultural R&D and innovation that enable agricultural emissions to diminish in the coming decades will be crucial to lessen the cost of achieving carbon neutrality. On balance, the more likely scenarios reveal the real cost of achieving carbon neutrality will not greatly increase. The cost of achieving carbon neutrality under the various scenarios is raised by an additional AUD22 million to AUD100 million per annum in constant 2020 dollar terms. This magnitude of cost increase is very small relative to the region's gross value of agricultural production that is regularly greater than AUD10 billion.

**Keywords:** carbon neutral; climate change; agriculture; emissions; reforestation

## 1. Introduction

Reducing greenhouse emissions is a popular sentiment, often voiced by industry organisations and governments. In Australia, key agricultural organisations have announced plans and commitments to achieve carbon neutrality [1–4]. Various state governments have passed legislation to deliver net-zero emissions by 2050 or earlier [5], and Australia's federal government, as a signatory to COP26, is committed to achieving net-zero greenhouse gas emissions by 2050.

The announced plans, actions, and aspirations for Australian agriculture to be carbon neutral means Australian agriculture now faces two structural challenges: lessening its net emissions whilst responding to an adverse trend in its climate. A wide-ranging review of climatic changes in Australia [6] reveals trends of rising temperatures, changing rainfall patterns, more extreme weather events, increasing ocean temperatures, and sea-level rise [7]. Agricultural production is being climatically more frequently adversely affected, particularly in southern regions of Australia [8,9], and human health is being negatively affected [10]. Various climate models project that these changes will continue and worsen [11–13]. It is against this backdrop of a challenging, potentially more adverse environment for agricultural production that Australian farmers currently are being encouraged to become carbon neutral.

Very little research in Australia focuses simultaneously on investigating the achievement of carbon neutrality against the backdrop of climate change. Often the focus of much climate change research is the consideration of how farmers could or should adapt to projected climate change [8,14–18] with no consideration of the additional need to achieve carbon neutrality. Another branch of the literature investigates abatement or mitigation activity in agricultural regions whereby farmers derive offset or carbon credit benefits via

activities such as increasing soil carbon levels [19–22], reforestation, or revegetation [23–25]. Often these contractual investments stretch over several decades, yet there are few longitudinal studies of abatement or offset activities in Australia that factor in the impact of climate change. A few studies do examine, mostly at the farm-level, the trade-off between farm profit and emission levels [26–28], but the cost to an Australian farm business or farming region of becoming carbon neutral against the backdrop of a changing climate is not reported in the literature, as far as the authors are aware.

Accordingly, to fill this gap in the literature, this study examines how climate change will affect the affordability of achieving carbon neutrality in south-western Australia; an important agricultural region in Australia in which the most frequently applied method of sequestering carbon is reforestation. To set the scene for this study, the next sub-section describes the study region and draws on Kingwell [29] to give an overview of the region's agricultural emissions, the categories of emissions, and the current trajectory of emissions. An online appendix provides technical information on Australian agriculture as a source of emissions, and reforestation as a source of emissions abatement. Then follows an outline of what is known about the likely impacts of projected climate change on agricultural production and tree production in the study region. A range of scenarios regarding agricultural emissions and impacts of climate change on tree growth and tree survival are presented. These scenarios are embedded in a spatial linear programming model that reveals how agricultural carbon neutrality via reforestation in the study region can be achieved at the least cost. The modelling results are presented and discussed. Finally, concluding remarks are made regarding the cost of achieving carbon neutrality via reforestation in the study region.

### 1.1. Study Region

The south-west of Australia is principally devoted to agricultural production. Farm size in the region is negatively correlated with annual rainfall, with large crop dominant farms being located on the drier inland edges of the region. In these marginal fringes, annual rainfall is under 300 mm and farm size is often over 10,000 hectares. Moving from the drier inland edges towards the coast sees increases in annual rainfall and reductions in farm size. In the far south-west, near the coast, annual rainfall is as much as 1000 mm and small mixed enterprise farms under 500 hectares are dominant. The south-west of Australia (Figure 1) generates almost 40% of Australia's winter crop production and supports 20% of the nation's sheep flock; and annually generates over AUD10 billion of agricultural production.

Agricultural production in the region is dominated by grain and sheep production, with the principal grains grown to be wheat, barley, and canola. Usually, around 4.9 million hectares are planted to wheat, 1.6 million hectares to barley and 1.4 million hectares to canola, and 0.7 million hectares to other minor crops. The region's current sheep population is around 14 million sheep. By international comparison, farms are large in area, typically more than 5000 hectares, yet are highly labour efficient, being run with fewer than four full-time labour equivalents, and are mostly owned and operated by farm families rather than large corporate entities. Over 90% of the region's production of major grains, sheepmeat (including live sheep exports), and wool are exported; so international commodity prices importantly influence farmers' crop and enterprise selections.

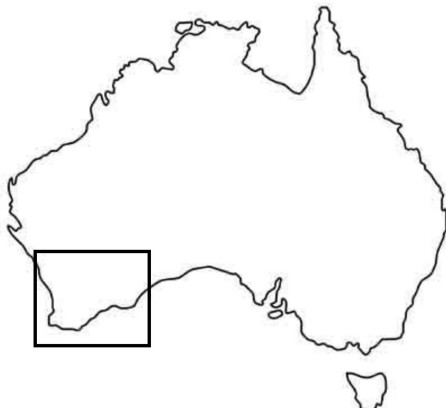

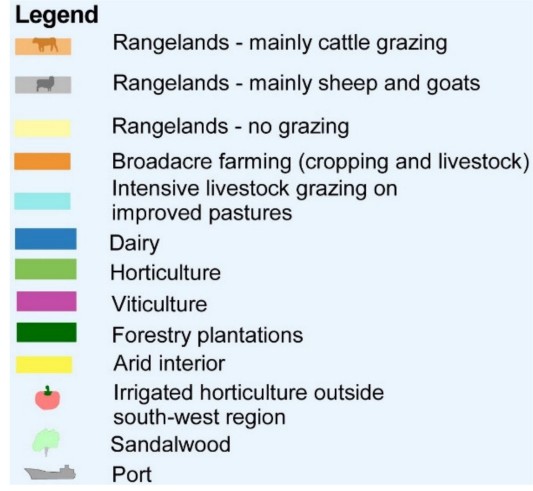
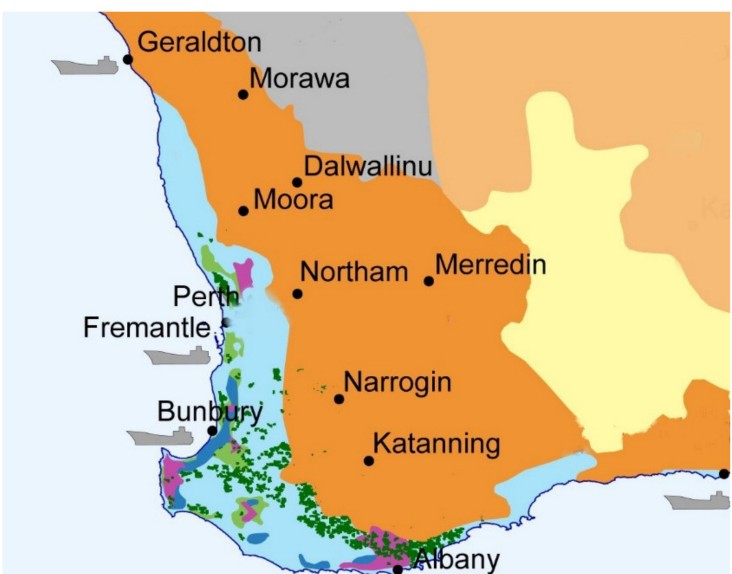

**Figure 1.** Land use in South-West of Australia. Source: Based on a DPIRD map available at: https://researchlbrary.agric.wa.gov.au/gis_maps/10/ (accessed on 12 November 2021).

### 1.2. Spatial Patterns of Emissions within the Region

The region's spatial pattern of emissions highlights those parts of the region that are the main sources of emissions and therefore identifies those businesses or industries liable to face the greatest challenges to achieving carbon neutrality. Spatial variability characterises the region's agricultural emissions [29]. The highest emitting shires are predominately wherever large numbers of livestock reside, and wherever higher crop input farming systems operate; and where the shire area is unusually large (Figure 2). The lowest emitting shires are mainly in the central wheatbelt, where the sheep population has greatly reduced and where shires are small in area. The swing away from sheep production since 1990, triggered by the collapse of the Reserve Price Scheme for wool in 1991 [30] and aided by subsequent productivity gain in cropping [31] has seen many farm businesses increase their crop dominance.

Future agricultural emissions in the study region will mostly depend on changes to cattle and sheep populations and how unfolding climate change may affect agricultural production and farm management. Since 2015 the sheep population in WA has stabilised whilst the WA cattle population has continued to decline from 2.24 million head in 2015 to 1.88 million in 2019. However, high prices for sheep meat and beef in recent years, plus planned major production investments in beef cattle production could see an increase in cattle and sheep numbers, and so enteric emissions could increase relative to levels observed in 2015 (Figure 2).

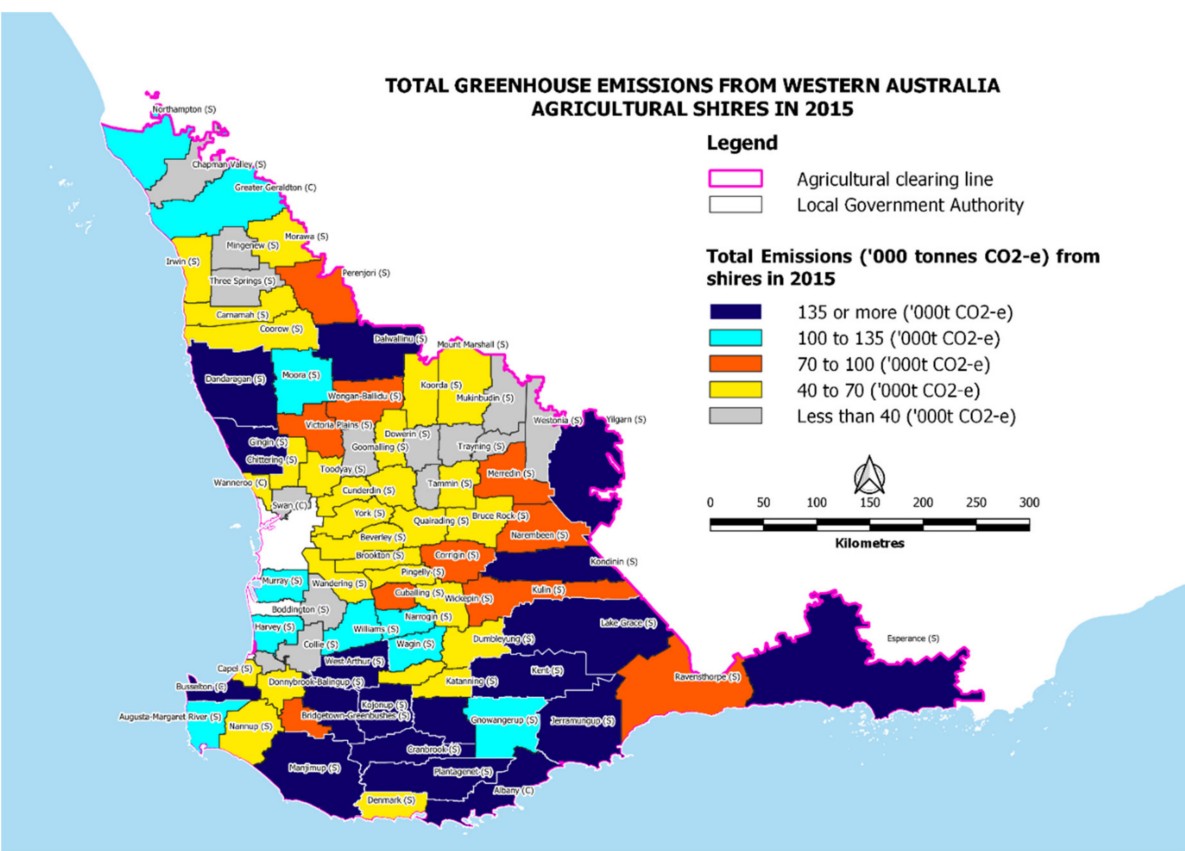

**Figure 2.** Shire emissions in 2015. Source: Kingwell [29].

### 1.3. Agricultural Impacts of Projected Climate Change

Climatic changes projected for Australia which are relevant for agriculture include trends of rising temperatures, changing rainfall patterns, and more extreme weather events [6,32–35]. For the south-west of Australia, both maximum and minimum temperatures are expected to rise, resulting in an average increase of between 0.6 °C and 1.5 °C by 2030. The intensity and incidence of severe weather events are also projected to increase over the coming decades. An increase in the number of dry days is also expected, which adds to the risk of wildfire.

Increased temperatures may change the locations where crops can be grown, and elevated $CO_2$ levels could affect crop growth and grain yield. Sudmeyer et al. [36] report climate projections for south-western Australia, noting the average annual temperature to increase by 1.1–2.7 °C and 2.6–5.1 °C by the end of the century under intermediate- and high-emission scenarios, respectively. Drawing on 18 global circulation models, Anwar et al. [9] report similar temperature projections for the region. Sudmeyer et al. [36] report that annual rainfall in the region is projected to decline by 12% by the end of the century (median values) for an intermediate emission scenario, and by 18% (median values), respectively, for a high emission scenario. Anwar et al. indicate average rainfall is projected to decrease by 16% in 2060 relative to a 1961–2010 baseline.

Figure 3 is a useful visual summary of the climate projections for the study region under an assumption of continued high levels of global GHG emissions (i.e., Representative Concentration Pathway (RCP 8.5)). Figure 3 draws on median values from 66 GCM model runs, using RCP 8.5 as the global emissions backdrop. Increased temperatures during grain-filling and during peak pasture production in spring (September to November) are destined to become problematic in the northern parts of the study region. The climate simulations were fairly consistent and are in accord with the most recent findings of Collins

and Chenu [37]. Reductions in rainfall will affect much of the region but could be especially problematic in the already drier eastern margins of the study region.

Agriculture in the region is almost completely rainfed, so the outlook of declining rainfall is likely to be the dominant-negative influence on agricultural production. Drawing on crop simulation models Anwar et al. indicate that the median total crop biomass of lupin, canola, and field pea crops could be about 10% to 40% less in the future climate relative to 1961–2010. Overall, these researchers identify that the impact of climate change on broadacre crops in south-western Australia will be increasingly detrimental towards 2090, resulting in potential yield losses reaching 42% for some crops. Similarly, Sudmeyer et al. [36] conclude that broadacre crop yields will be most affected by changes in rainfall, particularly the timing of rainfall, despite increased $CO_2$ concentration improving plant water use efficiency. Crop and pasture yields are projected to decline in the drier eastern and northern areas and remain largely unchanged or increase in wetter western and southern areas of the study region (Figure 1), especially where waterlogging was previously a problem. The plant available water capacity of the soil will become increasingly important to growth, so yield declines are likely to be greater on clay soils compared to sands in eastern areas of the region, as reported by Ludwig and Asseng [38]. Higher temperatures, and to a lesser extent declining rainfall, will hasten crop development times and reduce the flowering and grain-filling periods. Heat during grain-filling is likely to be increasingly problematic, and the production risks associated with climate variability will increase most in drier agricultural areas. The impact of reduced rainfall on grain yield is more severe on clay soils which hold more water in the topsoil and therefore lose more water to evaporation than sandy soils [39]. At lower rainfall, less water reaches deeper soil layers in clay than in sandy soils, increasing the risk of terminal drought on clay soils.

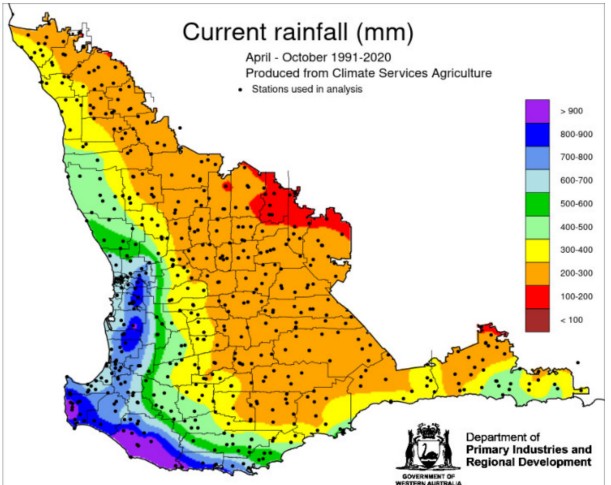 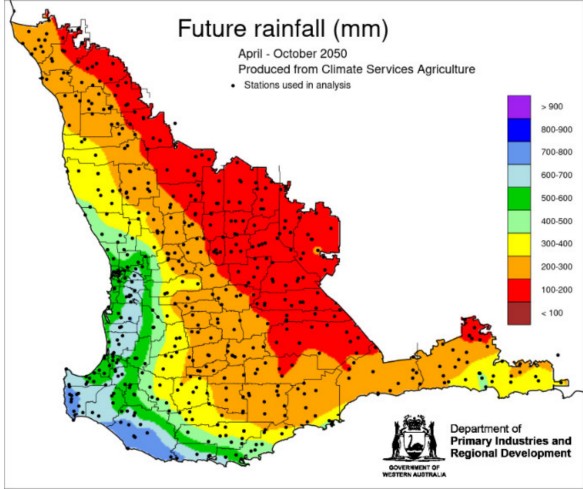

**Figure 3.** *Cont.*

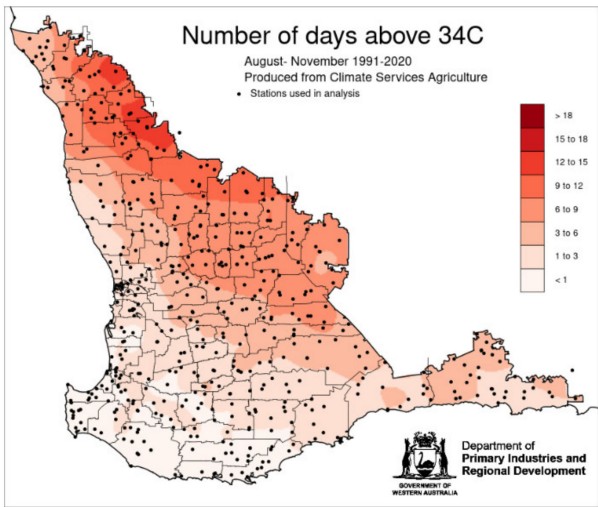 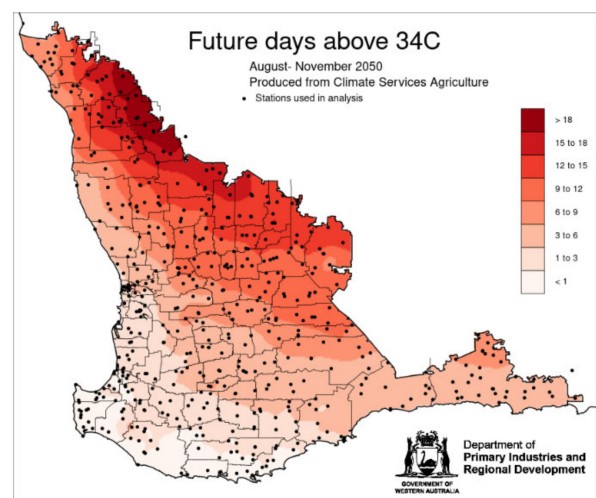

**Figure 3.** Climate maps for the study region. Source: DPIRD [40].

Baldock et al. [41] point out that emissions of greenhouse gases from soils are sensitive to soil temperature and water content, and so climate change may impact significantly on future emissions from soils. These authors comment that under dryland agriculture where water availability limits potential carbon capture by plants, a drier and warmer climate will likely reduce potential plant growth and restrict any potential increase in soil carbon. Moreover, changes in climate will also affect the magnitude of soil organic carbon loss from agricultural systems. Drying conditions reduce rates of soil organic carbon decomposition; but increasing temperatures increase rates of soil organic carbon decomposition [42,43]. Projected climate change impacts on soil organic carbon in south-western Australia suggest the net effect of changes in soil organic carbon on atmospheric $CO_2$ loading over the next few decades is overall likely to be small [44].

Yield predictions for the study region under projected climate change, as generated by crop simulation models, point to a decline in crop and pasture yields. However, these models usually exclude the offsetting benefits generated by technological and biological innovation. This limitation underlies the comments of Asseng and Pannell [8], who show that despite the twentieth-century changes in rainfall, temperature, and atmospheric $CO_2$ concentration in south-western Australia, no decline in wheat yields has been observed. Changes in agricultural technology and farming systems have had larger offsetting impacts, enabling water use efficiency to increase markedly. These authors boldly state that there is no scientific or economic justification for any immediate actions by farmers to adapt to long-term climate change in the Western Australian wheat-belt, beyond normal responses to short-term variations in weather. These researchers conclude that the most important policy response is research and development to enable farmers to continue their adaptation to climate change.

Updated evidence supports the view of Asseng and Pannell [8] (see Figure 4). However, it could be that the nature and magnitude of climate change in coming decades do eventually lessen productivity gains. Biologists and economists well know that response functions are rarely linear, and tipping points and points of discontinuity can arise. Quiggin and Horowitz [45], for example, point out that damages associated with climate change are a convex function of the rate of warming. Hence, as further warming occurs in the coming decades their observations suggest the level of damage will increase in a curvilinear fashion, increasingly testing the ingenuity of agricultural scientists, technologists, and farmers.

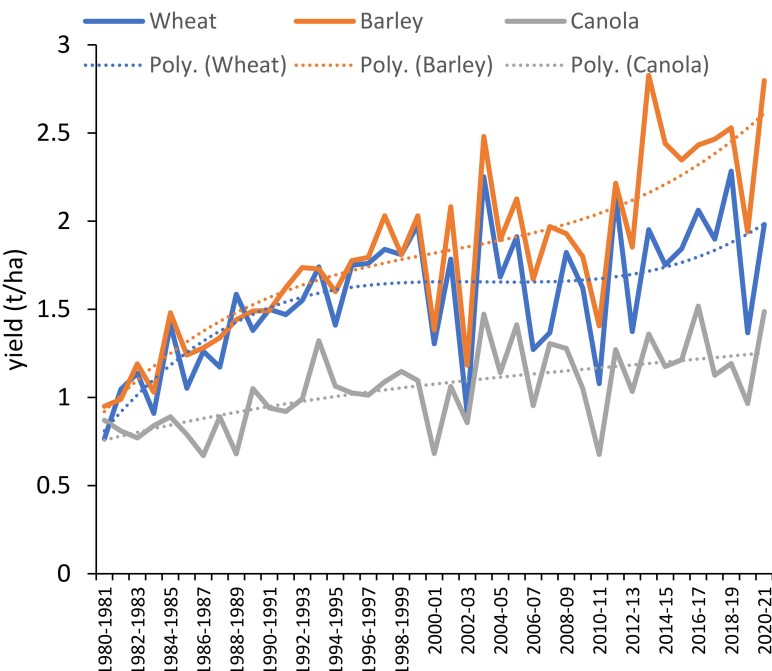

**Figure 4.** Wheat, barley and canola yields in south-western Australia: 1980 to 2020.

Turning to farm animals, projected climate change in the region will increase the number of days each year that livestock experience heat stress. Animals with heat stress have reduced appetite and are less likely to breed, resulting in productivity losses. Animal welfare considerations already affect animal production in the region [46], and any further heightened concerns are likely to alter animal production and transport systems to protect animals against extreme heat.

### 1.4. Options to Reduce Agricultural Emissions

To reduce agricultural emissions requires some combination of lowering emissions at source and/or using agricultural lands for sequestration. For mainstream extensive agriculture in southern Australia where adverse climate change is projected, emission reduction thus far has occurred principally through land use change involving a switch away from sheep production towards crop production. Sheep, being a ruminant, produce methane that is a particularly damaging greenhouse gas regarding its global warming potential. Hence, reductions in the sheep flock have importantly contributed to emissions reductions in south-western Australia [29].

Climate projections point towards pasture production being increasingly constrained by a decline in annual rainfall and increasingly hotter summers. As a result, a downward trend in a farm's carrying capacity will restrict the number of sheep that can be extensively grazed. Hence, adverse climate change that limits sheep numbers may contribute to a climate-induced reduction in sheep emissions that are the region's principal source of emissions.

Reduction in emissions is also feasible via sequestration that, in some situations, can involve improved levels of soil carbon [47,48], although as outlined by Baldock et al. [41], a drier and warmer climate will likely restrict any potential increase in soil carbon. Sequestration is also possible via agroforestry [49–51], re-vegetation and reforestation [52]; and avoided land clearing [53].

Reforestation and revegetation are by far the main abatement investments in Australia [54]. Vegetation projects are almost 70% of abatement projects that have been approved and funded by the Clean Energy Regulator in Australia. Accordingly, this study focuses on how agricultural carbon neutrality might be achieved via reforestation, where the challenge is to cost-effectively provide sequestration services that reduce net emis-

sions from agricultural activity to the point of carbon neutrality; under the challenge of a changing climate.

The emission accounting framework used in Australia and technical background on how agriculture is both a source and sink [55,56] of greenhouse gas emissions is outlined in Appendix A. That framework underpins scenarios of future emissions from agriculture in the south-west of Australia. Future agricultural emissions and reforestation abatement depend on a range of factors such as the magnitude and nature of unfolding climate change and the ability of agricultural scientists, technologists, and farmers to combat that adverse climate.

This paper uses scenario analysis to portray agricultural profitability and feasible emission and abatement activities likely to apply in 2050 and thereby reveals the plausible cost of achieving carbon neutrality via on-farm reforestation. For each scenario, the means of achieving carbon neutrality at least cost is described via a spatial linear programming model, described later.

The contribution of this study is that it identifies how agricultural carbon neutrality can be achieved via reforestation, in the least cost way in a key farming region of Australia under a changing climate, against the backdrop of likely social and political restrictions on the reforestation of farmland. The rationale for these land use restrictions is discussed later.

## 2. Materials and Methods

### 2.1. Scenarios for Agricultural Emissions and Abatement

Uncertainty surrounds possible trajectories of agricultural emissions in the study region, with many factors other than climate change affecting future emissions. For example, relative changes in the future prices of wheat, sheep meat, wool, canola and barley, and changes in the prices of their respective production inputs influence the enterprise mix on farms in the study region; as will technological change that underpins the production of various farm commodities. Some enterprises, such as sheep and wool production generate far greater emissions per hectare than cereal cropping enterprises. Hence, the enterprise mix on farms across the region and their associated technologies of production in the coming decades, mostly will affect emission levels.

Some emerging activities suggest there is a likelihood of emissions in the region tending to decrease. For example, effective anti-methanogenic feed supplements [57,58] are being developed and may be widely adopted, helping lower emissions from ruminant animals in the region; sheep, beef, and dairy cattle. Increasingly energy-efficient farm practices [59] and tailored variable rate technologies [60,61] are likely to further limit agricultural emissions. Honan et al. [62] reviewed the range of feed additives likely to reduce enteric methane production. The most efficacious was 3-nitroxypropanol (3NOP) which generated no apparent adverse effects on the animal or its subsequent food product. Reductions in methane output in cattle of around 50% have been recorded in various studies. However, further technical appraisals and regulatory approvals are required before this product could be commercially available. Phelan [63] describes masks fitted to the faces of dairy cattle. These lightweight masks capture and nullify the methane cattle breathe out. However, in the study region, the main animals in the region by far are sheep rather than beef or dairy cattle.

Kingwell [29] presents evidence that emissions in the region have lessened since the mid-1990s, principally due to a decline in the sheep population. He also shows that whenever a poor production year occurs, emissions are reduced. The previously described climate projections for the region suggest a greater likelihood of drought conditions that are known to lessen farm emissions and reduce sheep numbers. Hence, the more plausible scenarios for emissions beyond 2015 are for emissions to decline. Accordingly, a few emission scenarios are considered; with emissions in 2050 being unchanged, or reduced by 5%, 10%, or 20% relative to levels observed in 2015.

*2.2. Factors Affecting the Cost of Carbon Neutrality for the Region's Agriculture*

Just as emission trajectories are conditional on many influences, so the cost of achieving carbon neutrality by 2050 via reforestation is conditional on several factors. The likely range of values of each factor is described below.

(i).    the nature of emission trajectories

As previously discussed, emissions in 2050 compared to known emissions in 2015 are plausibly considered to be either unchanged, or reduced by 5%, 10%, or 20%. Reductions in agricultural emissions towards 2050 potentially lessen the area of farmland that needs to be committed to reforestation for carbon abatement to achieve carbon neutrality.

(ii).   the impact of climate change on tree survival and tree growth

It is well-established that tree growth in the region is largely a product of annual rainfall [64–67]. In their study of tree growth performance and survival in the study region, Spencer et al. [67] report: "The most likely reason for the slower growth rates of many planting configurations is the lack of available water. In the Western Australian wheatbelt, the annual potential evaporation (PET) can be up to five-fold the annual rainfall . . . water has been shown to be a major limiting resource."

Most studies that have examined recent patterns of tree growth in agricultural regions have focused on plantings of a few species among the over 400 mallee species that are native eucalypts [67–70]. Their tree growth patterns follow a Gompertz curve, and un-harvested stands of these trees usually experience maximum carbon storage four decades after planting. Tree growth survival often depends on initial growing conditions, with Spencer et al. [70] reporting tree survival rates for plantings in 2000 of between 69 to 94%.

Drawing on 18 global circulation models Anwar et al. [9] indicate the average rainfall in the study region is projected to decrease by 16% in 2060, relative to a 1961–2010 baseline. These projected declines in rainfall are likely to lessen tree growth. In our analysis we assume, under future climate, two different tree growth scenarios towards 2050 in which tree growth is either 10% or 15% less than in recent climate.

(iii).  the impact of climate change on the risk of catastrophic wildfire

The IPCC [12] outline how concurrent hot and dry conditions amplify conditions that promote wildfires, and the IPCC report a long-term trend towards more dangerous weather conditions for bushfires in many regions of Australia. The increased fire risk in southern Australia [7,71] lessens the efficacy of reforestation for achieving carbon neutrality. Worse is that an increased frequency of wildfires adds to atmospheric greenhouse gases [72].

Currently, carbon forestry projects supported by the Australian government's Emissions Reduction Fund (ERF) must comply with permanence rules that require that the carbon stocks be retained for 100 years, although the ERF has introduced an optional permanence period of 25 years. However, a project proponent using the 25-year period receives 20% fewer Australian Carbon Credit Units (ACCUs) [53]. In approved carbon forestry projects no account is taken of any additional benefits to soil carbon under a switch into permanent forestry, although these benefits are likely to be small [73], and no account is taken of bushfires that remove sequestered carbon, other than the 5% risk-of-reversal buffer. Noting the IPCC's [12] finding that the risk of wildfire will increase in many regions of Australia, including the study region, the 5% risk-of-reversal buffer may be inadequate. Accordingly, in our analysis, we increase that buffer to 7.5%.

(iv).   socio-political constraints on conversion of farmland into permanent forests

Kingwell [29] mentions that in the study region reforestation of farmland is often associated with social conflict. Williams [74] surveyed community attitudes to wood plantations in the study region and found respondents valued agricultural land use higher than plantation forestry. Moreover, many people believed plantations offered most benefits primarily to timber company shareholders and created limited regional economic benefits. The collapse of many forestry-managed investment schemes in the study region in the

2000s [75] has cemented these oppositional views. Accordingly, most local governments in the study region are unlikely to support widespread reforestation of farmland in their locality. The assumption made in this study is that no more than 25% of the farmland in any local government area in the study region would ever be allowed to switch out of farming into reforestation.

(v).   the real value of farmland in 2050

The cost of switching land out of farming into carbon forestry does depend on the value of farmland in 2050. In recent decades farmland prices in the study region have grown strongly. RuralBank [76] reports that farmland prices in the study region have increased over the previous two decades at an annual average growth rate of 5.4%. Over the same period price inflation has increased at only 2.5% per annum [77]. Hence, real growth in farmland prices in the study region has been 2.9% per annum over two decades.

Whether the observed real growth in farmland prices continues towards 2050 depends on several factors, including the future agricultural impacts of climate change in the region, particularly regarding crops yields, as farming systems throughout the study region are crop dominant. Climate change projections point to a projected decline in rainfall that crop simulation models indicate will generate yield declines. However, we side with the views of Asseng and Pannell [8] regarding the vital role of innovation in allowing crop yields to further increase. In this study, we assume that crop yields in 2050, relative to trend yields in 2015, when combined with input innovation will maintain growth in the real value of farmland, albeit at a lesser rate. Specifically, we assume that the real annual growth in farmland prices in the study region will be 2.0%.

(vi).  the magnitude of price premia in 2050 for regional farm products able to be branded as carbon-neutral

During the next decade, an increasing array and size of markets for carbon neutral farm products are likely to emerge. However, by 2050 we assume that the general expectation from consumers will be that most farm products sold will be carbon neutral, and furthermore, there will be no price premium for the carbon neutrality status of an agricultural product. Rather the social licence to operate as a seller of farm products will include the requirement to be carbon neutral. Hence, in this study, we assume by 2050 there are no price premia for carbon neutral farm products. Accordingly, farm profits and farmland prices are assumed to not be supported by price premia for carbon neutral farm production.

(vii). the magnitude of co-payments in 2050 for other complementary environmental services generated by reforested areas

To further encourage environmental plantings in farmland regions, especially on soils or in landscapes characterised by low agricultural productivity we assume that governments will eventually offer farmers payments for the complementary environmental services generated by reforestation, especially mixed species revegetation. The co-benefits include wildlife habitat, reduced salinisation, shade and shelter for farm animals, and enhanced biodiversity. We assume that the stream of these co-benefits reduces the present value cost of establishing areas of reforestation by 20%.

### 2.3. Modelling the Cost of Carbon Neutrality Given Climate Change

The cost of achieving carbon neutrality, based on reforestation, for the study region under different climate change and factor scenarios, can be couched as a steady-state linear programming (LP) land allocation problem. For each scenario, the cost of required abatement in 2050 can be minimised. In each scenario, this minimisation objective is subject to various constraints, including social or political constraints on how much farmland in each shire can be switched into reforestation. These likely political or social restrictions are represented by the proportion p of each shire's land being made available for sequestration

activity. The LP problem for each scenario can be stated as a steady-state problem in 2050 and mathematically is:

$$Min \sum_{i=1}^{n} l_i C_i$$

subject to:

$$(l_i + a_i) = T_i \text{ for each shire } i = 1, 2, \dots, n \tag{1}$$

$$l_i \leq pT_i \text{ for each shire } i = 1, 2, \dots, n \tag{2}$$

$$\sum_{i=1}^{n} l_i S_i = \sum_{i=1}^{n} a_i E_i \tag{3}$$

$$l_i \geq 0$$

$$S_i \geq 0$$

$$E_i \geq 0$$

where

$l_i$ is the land (hectares) allocated for sequestration in shire $i$

$a_i$ is the land (hectares) allocated for agriculture in shire $i$

$C_i$ is the annual cost (in constant 2020 AUD) of sequestering a tonne of $CO_2$-e per hectare in shire $i$ in each scenario

$T_i$ is the total area of land (hectares) available for agriculture and sequestration in shire $i$

$p$ is the proportion of land available for agriculture and sequestration in shire $i$ that is legally able to be devoted to sequestration activity

$S_i$ is the tonnes of $CO_2$-e sequestered annually per hectare in each scenario on land allocated for sequestration in shire $i$

$E_i$ is the emissions (tonnes of $CO_2$-e per hectare) in each scenario generated on agricultural land in shire $i$

$n$ is the total number of shires in the study region ($n = 80$).

Equation (1) describes how the farmland in each shire must be allocated either to agriculture and/or reforestation. Equation (2) captures the political or social constraint that up to a proportion p of the land available for farming and reforestation in each shire can be reforested. Equation (3) specifies that, in each scenario, the region's annual sequestration should equate to the region's annual agricultural emissions. The other constraint equations are non-negativity conditions that typify most LP problems.

The LP model described above applies to the scenarios (see Table 1) mentioned in the previous points (i) to (vii). The various scenarios modelled reflect a plausible range of agricultural emissions in 2050, the likely climate-induced reductions in tree growth, a greater incidence of wildfire, a lesser rate of growth in farmland real prices, and the provision of co-benefit payments for reforestation.

A few other studies have applied LP to land allocation problems related to emissions reduction (Huang et al. [78], Smith et al. [79]). Huang et al., for example, used LP to illustrate how land use planning could help minimize carbon emissions in an administrative region of China.

**Table 1.** Key characteristics of optimal land use in the region to achieve regional carbon neutrality under various scenarios.

| Scenario | Description | Region's Annual Emissions (kt $CO_2$-e) | Reforestation Area in the Region that Achieves Carbon Neutrality (m ha) | Share of the Region's Farmland Switched into Reforestation (%) | Share of the Region's Population of Shires that Invests in Reforestation (%) | Annual Cost for the Region to Achieve Carbon Neutrality via Reforestation ($m) |
|---|---|---|---|---|---|---|
| 1 | Base case | 7425 | 1.333 | 8.6 | 26.2 | 216.0 |
| 2 | Emission levels unchanged, 10% decline in tree growth | 7425 | 1.664 | 9.3 | 38.8 | 297.4 |
| 3 | Emission levels unchanged, 15% decline in tree growth | 7425 | 1.766 | 11.4 | 41.3 | 316.6 |
| 4 | Emission levels decline 5%, 10% decline in tree growth | 7054 | 1.572 | 10.2 | 36.3 | 282.3 |
| 5 | Emission levels decline 5%, 15% decline in tree growth | 7054 | 1.691 | 9.1 | 38.8 | 300.8 |
| 6 | Emission levels decline 10%, 10% decline in tree growth | 6683 | 1.490 | 10.4 | 32.5 | 267.5 |
| 7 | Emission levels decline 10%, 15% decline in tree growth | 6683 | 1.605 | 10.8 | 37.5 | 284.7 |
| 8 | Emission levels decline 20%, 10% decline in tree growth | 5940 | 1.308 | 8.5 | 28.8 | 237.5 |
| 9 | Emission levels decline 20%, 15% decline in tree growth | 5940 | 1.400 | 9.0 | 31.2 | 253.0 |

## 3. Results and Discussion

The base case in Table 1, scenario 1, refers to the current situation in 2020 and outlines the steady-state cost of ensuring carbon neutrality for the region via reforestation. By committing 8.6% of the region's farmland into reforestation, the resulting annual abatement equates to the annual agricultural emissions from the region, assuming steady-state conditions. The annual cost of achieving ongoing carbon neutrality is the foregone value of agricultural activity on reforested land and the annualised expense of planting, maintaining, and monitoring the reforested areas. These annual costs are AUD216 million. Kingwell [29] shows that for farmers in the region to self-finance the region's carbon neutrality via reforestation activity, would cause current aggregate annual farm business profit in the region to decline by 15%.

Results listed in Table 1 reveal that the impact of climate change in the region will increase the cost of achieving carbon via reforestation as the projected decline in rainfall, combined with further warming, will lessen tree growth, and a greater area of reforestation will be required to offset agricultural emissions. If technological innovation however enables agricultural emissions in the region to lessen, then despite the impacts of climate change on tree growth, the real cost of achieving carbon neutrality via reforestation is projected to increase little. For example, if agricultural emissions reduce by 20% in 2050 relative to levels in 2020, yet climate change lessens tree biomass production by 10% (i.e., scenario 8), then the cost of achieving regional carbon neutrality (in 2020 dollar terms) only increases by AUD21.5 million which is a tiny amount given that the region's gross value of agricultural production is usually over AUD10 billion. Noting that methane emissions are the main source of emissions in the region [29], methane-reducing innovations as described in [57,58,62,63] could make an important contribution to lowering the cost of achieving carbon neutrality in the presence or absence of climate change.

These findings emphasize the important role agricultural innovation needs to play in the coming decades. Not only must agricultural R&D enhance farm productivity amid the environmental challenge of a drying and warming environment (Figure 3) but the technologies and practices that underpin agricultural production in coming decades must generate fewer emissions. If agricultural profitability is not maintained or if emissions cannot be reduced, then the cost of achieving carbon neutrality for agriculture in the region, via reforestation, will only increase, and farm profits in the region will be more substantially reduced via a requirement to be carbon neutral.

The costliest scenario for achieving carbon neutrality is where there is no decline in agricultural emissions towards 2050, despite the best endeavours of technology innovators, and yet projected climate change reduces tree growth by 15%, and the risk of wildfire increases (i.e., scenario 3). In this case, a larger area of farmland is required to be reforested with agricultural profits on the enlarged area of reforestation being foregone. The cost of achieving carbon neutrality, relative to the base case, increases by AUD100.6 million or by almost 47%.

In most scenarios listed in Table 1, between 8.5% and 11.4% of the area of farmland in the study region needs to be switched out of agriculture into reforestation to achieve sufficient sequestration to achieve carbon neutrality. The shires in which reforestation occurs do change (Figure 5), dependent on the scenario, with the proportion of the population of shires engaging in reforestation under climate change, ranging from 29 to 41%. Almost irrespective of the scenario, however, the shires that are the main sources of reforestation are often larger shires in the far south and east of the agricultural region (Figure 5). These shires tend to be selected as the least-cost sources of sequestration for different reasons.

The southern shires selected to be main sources of reforestation typically are located in higher rainfall environments (see Figure 1) where tree growth is greater and where farmland prices are mostly influenced by the profitability of agriculture rather than the spillover effects of tourism, hobby-farming, holiday-making or urbanisation, as in other higher rainfall locations. These southern shires selected for reforestation often have more livestock in their farming systems and therefore higher levels of emissions (Figure 2). So, an added benefit of reforesting farmland in these shires is a greater decrease in emissions through the reduced availability of farmland to carry livestock. It is worth noting that any decline in animal numbers is unlikely to greatly effect on-farm prices for animals and animal products (mostly wool) as the principal markets for these commodities are price elastic export markets. For example, wool produced and exported from the study region is but one of several regions in Australia, New Zealand and South Africa that produce and export wool.

The eastern shires selected for reforestation (Figure 5) are in localities characterised by low rainfall (Figure 1) and constrained tree growth. Farmland in these eastern shires is historically among the most affordable to acquire [76] as the farms are located on the environmental margin for crop and sheep production. The cheapness of the land, even after acknowledging the limitations on tree growth, allows these shires to be cost-effective sources of sequestration.

In scenarios where the region's agricultural emissions do not diminish, yet tree growth reduces due to adverse climate change, and the risk of wildfire increases, then a large area of reforestation involving more shires is required. In these scenarios more farmland in many medium rainfall, southerly shires are reforested (for example, Figure 5, scenario (a)). When agricultural emissions do diminish, as in scenario (b) in Figure 5, then despite reduced tree growth, a lesser area of reforestation and fewer shires are selected to ensure carbon neutrality for the region.

Although the results are not presented here, if the socio-political constraint on the proportion of farmland in any shire that can be reforested is relaxed, then only a handful of mostly southerly shires are completely reforested and the cost of achieving carbon neutrality is reduced. In the worst-case scenario where agricultural emissions are unchanged, yet tree growth diminishes by 15%, then the cost of achieving carbon neutrality falls from AUD316.6 million (Table 1) to AUD272.3 million. In this case, only 9 shires (i.e., 11% of the shire population) are selected for reforestation when the socio-political constraint on reforestation is lifted. This finding reveals that honouring societal views about the need to restrict the proportion of farmland that can be reforested does increase the financial cost on the region's agricultural sector in its achievement of carbon neutrality via reforestation.

An important assumption in this analysis is that the most profitable use of land in the study region is for agriculture rather than reforestation. Another less plausible scenario not considered in this analysis is that growth in the real price of ACCUs (Australian Carbon

Credit Units) will be such that investors in carbon forestry could financially benefit from purchasing farmland for reforestation, despite the likelihood that climate change will lessen tree growth. To date, the carbon price in Australia (i.e., the value of an ACCU; see [54]) has not risen to any level that would entice switching of farmland into reforestation. In the study, regional farmland continues to be bought by entities who solely wish to continue to invest in agriculture.

A separate but important issue not considered in this study is that it might be more economically sound for agriculture in the study region not to invest in reforestation to offset agricultural emissions but rather to facilitate emission reductions in other sectors. For example, it may be preferable for farmland not to be reforested but rather for farms in relevant locations to offer up their farms for the joint use of agriculture and renewable energy generation. Emissions from coal or gas-powered electricity generators could be reduced through wind-powered or solar electricity generation on farmland. Before switching farmland into permanent forests, it may be technically and economically more feasible for farmers to first facilitate emission reductions in other sectors.

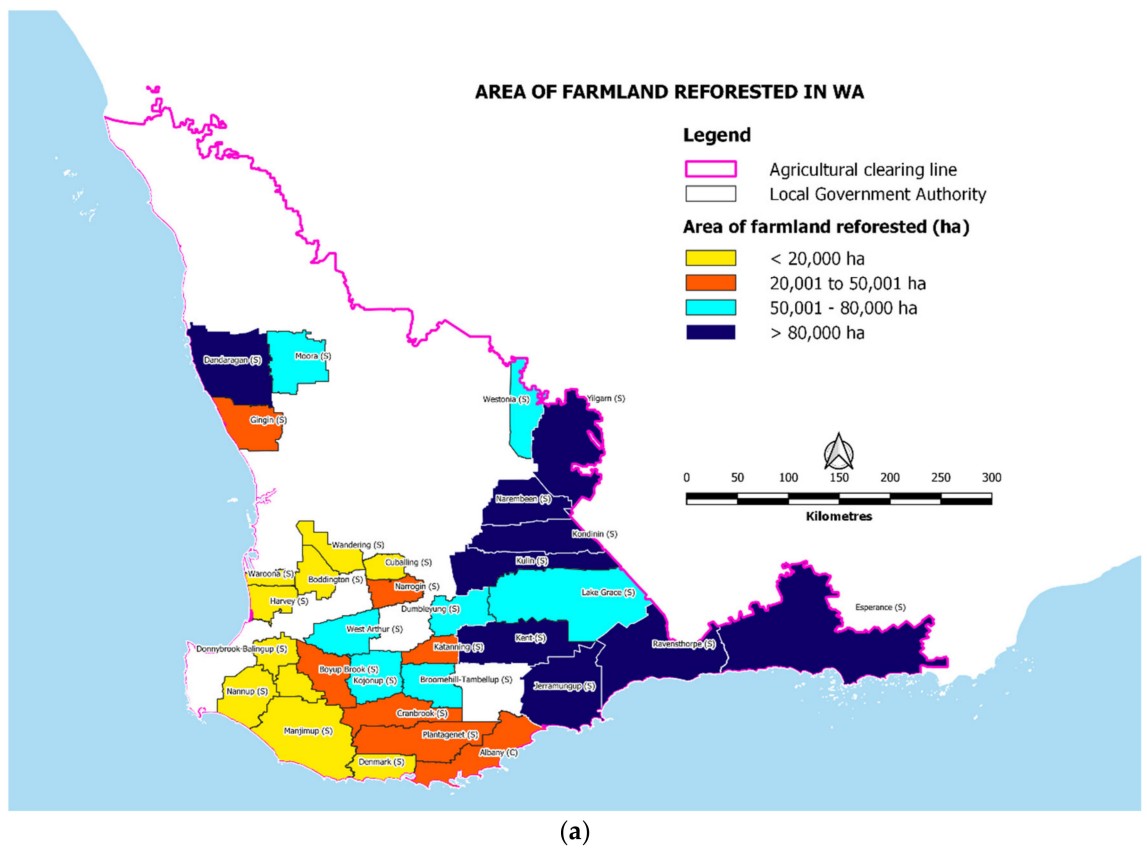

(**a**)

**Figure 5.** *Cont.*

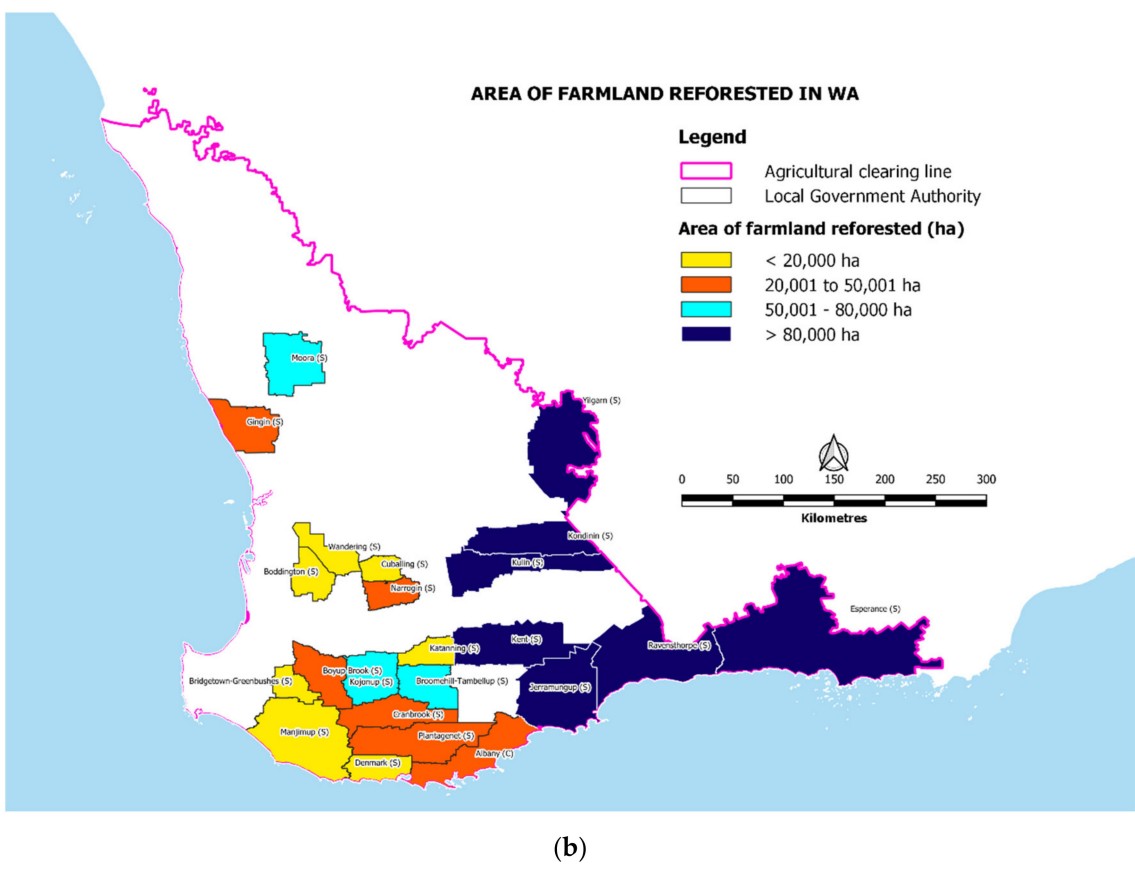

(**b**)

**Figure 5.** Area of farmland in various shires selected for reforestation under two scenarios; top (**a**) no decline in agricultural emissions towards 2050 and tree growth reduces by 15% and a greater risk of wildfire, and bottom (**b**) agricultural emissions decline by 20% towards 2050 and tree growth reduces by 10% and risk of wildfire increases.

## 4. Conclusions

In south-western Australia, projected adverse climate change is acknowledged to likely lessen tree growth and increase the risk of wildfire in reforested areas. By contrast, despite this likelihood of a more adverse climate, ongoing innovation in agriculture will likely lead to further constrained growth in agricultural production, whilst simultaneously enabling a reduction in agricultural emissions. The interplay of these consequences on tree production and agricultural emissions will affect the cost and feasibility of achieving agricultural carbon neutrality via reforestation of farmland.

Scenario analyses that consider different degrees of climate change impacts and different emission trajectories reveal that on balance, despite the likelihood of agricultural emissions being reduced in coming decades, the impact of projected adverse climate change on tree growth and tree survival means that the cost of achieving agricultural carbon neutrality via reforestation will increase in the study region of south-western Australia.

Reforestation becomes increasingly expensive for two main reasons. Firstly, trees store less carbon under projected adverse climate change, and the risk of wildfire increases, which means the sequestered carbon cannot be so securely stored. Secondly, real appreciation in farmland prices is likely to continue which increases the expense of using farmland for reforestation.

If agricultural R&D and innovation enable agricultural emissions to diminish in the study region, then despite the adverse impacts of climate change on tree growth and tree survival, the real cost of achieving carbon neutrality will not greatly increase. Under such a scenario where agricultural emissions fall by 20% towards 2050 and tree growth reduces by 10%, then the annual cost of achieving agricultural carbon neutrality via reforestation increases to AUD237 million in constant 2020 dollars from a base case cost of AUD216

million per annum. This magnitude of cost increase is very small relative to the region's gross value of agricultural production that is regularly greater than AUD10 billion.

**Funding:** This research received no external funding.

**Data Availability Statement:** The datasets that underpin the linear programming models in this paper are available from the author on reasonable request.

**Acknowledgments:** I wish to acknowledge the assistance of Mariana Parra Loza who generated the maps in Figures 4 and 5 from data provided by the author. I also wish to thank Amir Abadi for alerting me to recent literature on sequestration research in the study region.

**Conflicts of Interest:** The author declares no conflict of interest.

## Appendix A

Technical information on Australian agriculture as a source of emissions, and reforestation as a source of emissions abatement.

### Appendix A.1. Agricultural Sources of Greenhouse Gas Emissions

Greenhouse gases are released when biomass decays or is consumed or burnt [80]. Agricultural practices have increased these processes through the introduction of cropping and livestock systems. The primary greenhouse gases produced by agriculture are methane ($CH_4$) and nitrous oxide ($N_2O$) [81]. Methane and nitrous oxide have a greater Global Warming Potential (GWP) than carbon dioxide at 21 and 310 times respectively [82], with those values being altered to be 25 and 298 respectively in the IPCC Fourth Assessment Report and then changing again, starting in 2020–2021, in accordance with the terms of the Paris Agreement, to be 28 and 265 respectively [83]. Hence over the last decade the scientific consensus has indicated a heightened GWP for methane and a lesser GWP for nitrous oxide.

Agriculture is responsible for 85% of Australia's total nitrous oxide emissions primarily due to the application of nitrogenous fertilisers, cultivation of nitrogen fixing crops and pastures, and tillage of agricultural soils [84]. Agriculture is also responsible for 60% of total methane emissions [84]. Methane is released from the process of enteric fermentation in the digestive process of livestock, particularly in ruminants. In anaerobic conditions methane can also be produced from manure and this is particularly associated with intensive livestock industries. Nitrous oxide can be released from manure and urine on soil, but emissions are only significant in high rainfall areas [84].

A growing source of agricultural emissions is the soil amelioration practice of liming to increase soil pH on acidic soils and thereby improve plant growth. Incorporation of lime into acidic soils causes a chemical reaction that produces $CO_2$. In some states of Australia, particularly in Western Australia, are large areas of acidic or acidifying soils that benefit from periodic applications of lime. Umbers [85] notes that since the mid-2000s the percentage of the crop area limed in most grain-growing regions of Australia has increased from approximately 5% to around 25% in 2016. However, the rate of lime applied has remained fairly stable at under 2 tonnes per hectare. A further growing source of emissions is the propensity of grain farmers to apply more urea fertiliser to crops for various reasons including the planting of higher-yielding varieties that benefit from applications of urea, a greater role of canola in cropping programs with canola requiring higher rates of application of urea, plus a diminished role of leguminous pastures in farming systems that encourages farmers to replace their biological nitrogen with nitrogen from fertilisers.

Principally due to widespread drought in eastern Australia, the millennial drought and repeated drought in 2017 and 2018 that triggered culling of grazing animals, main sources of agricultural emissions, the total emissions from Australian agriculture have trended downwards since 2000. Agricultural emissions in 2018 were 8.4% less than in 2000 [83], already satisfying that sector's Kyoto Protocol requirements.

*Appendix A.2. Reforestation as a Carbon Sink*

Agriculture can reduce or offset its greenhouse gas emissions through reforestation or agroforestry that sequester carbon dioxide [20,52,84,86]. Articles 3.3 and 3.4 of the Australian ratified Kyoto Protocol allow for emission offsets through the sequestration of carbon. Article 3.3 covers reforestation and afforestation activities occurring after 1990, subject to the following conditions [87]:

i.   Land was cleared prior to 1990
ii.  Trees at a minimum height of 2 metres
iii. Forest crown cover of at least 20%
iv.  Forest area greater than 1 hectare
v.   Forest established by direct human methods

Reforestation and plantation-based sequestration activity under Article 3.3 is supported by the Emissions Reduction Fund (ERF). The Plantation Forestry Methodology Determination (also known as the ERF Plantation Forestry Method) covers the establishment of a new plantation forest, conversion of a short-rotation plantation to a long-rotation plantation, or maintenance of a pre-existing plantation forest that meets the eligibility requirements of the method. Projects approved by the Clean Energy Regulator generate Australian Carbon Credit Units (ACCUs) where each ACCU represents a tonne of carbon dioxide equivalent net abatement (through either emissions reductions or carbon sequestration) achieved by the eligible project. A new additional step in the project approval process is that the federal minister for agriculture, water and the environment may also assess if a proposed project could lead to an undesirable impact on agricultural production in the region in which the project would be located.

The ERF Plantation Forestry Method complements agroforestry activity permissible under the Carbon Farming Initiative (CFI). The CFI was a voluntary carbon abatement scheme that ran between September 2011 and December 2014 after which it was integrated with the Emissions Reduction Fund (ERF) such that an existing CFI project automatically became an ERF project. The regulatory burden for forestry sector participation in the ERF was eased in 2020, while recognising the need to ensure ERF forestry projects would not pose a cumulative adverse risk for water availability [88].

Several requirements must be satisfied before a forestry or reforestation project can be declared an 'eligible offsets project' including among other things that the project must comply with an approved methodology and the project proponent must report to the Regulator about the conduct of the project and the abatement achieved, with certain reports being prepared by a registered greenhouse and energy auditor. Importantly, the permanence rules require that the carbon stocks in sequestration projects be retained for 100 years, although the ERF has introduced an optional permanence period of 25 years. However, a project proponent using the 25-year period will receive 20% fewer ACCUs [52].

*Appendix A.3. Emissions Accounting*

Methods for reporting agricultural greenhouse gas emissions are stated in the National Greenhouse Accounts, including equation 3G_1 [89] for emissions from liming of agricultural soils and equation 3H_1 for emissions from applications of urea. Kingwell [29] outlines the methods for estimating agricultural shire emissions in Western Australia.

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
