# Peer review of "Making Agriculture Carbon Neutral Amid a Changing Climate: The Case of South-Western Australia"

_land, doi:10.3390/land10111259_

Round 1

Reviewer 1 Report

The manuscript reports on reforestation as a means to make agriculture in SW Australia carbon neutral by 2050. 

There is a base for a publishable manuscript but it would benefit from some additional detail as suggested below.

line 105 section 'Spatial patterns of emissions within the region' the key agricultural commodities are described.  can the authors provide details of the inputs that go into these e.g. nitrogen fertiliser, crop protection, machinery and the emissions associated with each to give an emissions profile.  then link this to the spatial map of Figure 2 based on the quantity of these commodities produced in each area.  this can then be linked to the predicted changes in climate in Figure 3.

line 177 'The plant available water capacity of the soil will become increasingly important to growth, so yield declines are likely to be greater on clay soils compared to sands in eastern areas of the region.' this needs explaining in terms of the size of soil particle and drainage potential of a given soil type.  does clay not potentially retain more water due to a lower drainage capacity? although the smaller particles retain a layer of non extractable moisture around each

line 188 'Moreover, changes in climate will also affect the magnitude of soil organic carbon loss from agricultural systems. Drying conditions reduce rates of soil organic carbon decomposition; but increasing temperatures increase rates of soil organic carbon decompo-190 sition [38,39].' needs additional detail as to the mechanism i.e. microorganisms not functioning where there are extremes in soil conditions i.e. extremely wet or dry

line 192 'Yield predictions for the study region under projected climate change, as generated by crop simulation models, point to a decline in crop and pasture yields.'  are these yield maps available spatially i.e. to allow the reader to see how things will change relative to the current situation in given parts of the region and allow the spatial targeting of adaptation strategies.

line 260 'This paper uses scenario analysis' what are the scenarios used specifically? those in Table 1?

line 280 'Some emerging activities suggest there is a likelihood of emissions in the region tending to decrease. For example, effective anti-methanogenic feed supplements [53,54] are being developed and may be widely adopted, helping lower emissions from ruminant animals in the region; sheep, beef and dairy cattle. Increasingly energy efficient farm practices [55] and tailored variable rate technologies [56,57] are likely to further limit agricultural emissions.' - can the authors provide details / figures of how much GHG reduction is attributed for each of these emerging technologies.

line 395 'The cost of achieving carbon neutrality, based on reforestation, for the study region' why is reforestation is the only abatement strategy considered?  what happens once tree growth in the reforested areas stop ie. attain equilibrium? i acknowledge this will be after 2050 (40 years after planting according to line 317) but is an important consideration as it is approx 2060 if planted now.  line 316 mentions 'native eucalypts'.  are there other species / species mixtures of greater value applicable in different areas? can the Gompertz curve be described with its implications for carbon sequestration? was a spatial analysis accounting for agricultural yield considered so the least productive areas of farmland could be targeted, this would minimise the risk of production displacement 

Reviewer 2 Report

    This study applies a very simple linear model that balances carbon sequestration and agricultural emissions. There is no spatial structure in the equation, so local ecology is not included. The model ignores carbon offsets, changes in technology, changes in crops or management that alter soil carbon, etc. A few of these are mentioned but they are not in the model. There is no equilibrium explored for the loss of livestock/rangeland. It should emphasize that this is a very simple back-of-the-envelope model.   This manuscript follows some of the previous paper (Kingwell 2021). However, there is really nothing at all I could find that really offers even a simple model like this, so I think the novelty outweighs the simplicity. very good idea, I just think it warrants more careful language recognizing the limits of such a simple model. I include some comments aimed at making the manuscript more readable.   Figure 1. You already have precip in Figure 3. Could Figure 1 instead be crop type, or land cover type?   line 144:  are most of the 18 climate models consistent for WA?  or is there considerable variability in their projections? You should note how consistent/inconsistent they are with one another. You mention median rainfall (but not temperature?) in the next sentence but I am unclear how useful that is. What would really make this interesting is to do some Monte Carlo trials, based around actual distributions of carbon, area, and costs, and see what the results show.   A significant amount of intro material is in the Materials and Methods section. I recommend starting the Methods section around line 270 (and moving the Study Region segment here).  The scenarios section is long; could this be modified to simply refer the reader to appropriate references?   Figure 3:  Is it OK to use figures produced elsewhere like this? Seems like a copyright issue, but maybe it's fine. Also needs panel labels (abcd).   Also figure 3: Heat and water stress are much more variable than the temperature maps let on. I would rather see water stress maps than 34C exceedence maps as the impacts are very crop-specific.   Sudmeyer et al 2016, and Asseng and Pannell 2013, are cited several times for WA.  There are several other studies  (e.g. Taylor et al 2018) that suggest wheat yields will decline -- can you address the discrepancies about WA agricultural projections?   Figure 4: the vertical axis says "Wheat yield" --- do you mean just "yield"?   Can you give an example of a linear programming land allocation applied to a somewhat similar situation, other than your previous (Kingwell 2021) paper?   Figure 5 is just converting the percentages to estimate future acreages. I think this is not really direct, because the model is essentially aspatial, but you are applying an aggregate trend to spatial data to kind of make your results spatial. If we take your extreme switch of 11.4%, the model indicates more reforestation in areas with lower future rainfall (from Figure 3 b). So I should expect more than 80k ha reforested even under drying/ higher fire conditions in the eastern part of the study area? Maybe it's economically balanced, but I'm not confident it's biophysically possible.    

Reviewer 3 Report

This excellent paper investigates agricultural emissions and reforestation potential for carbon sequestration in South Western Australia using a scenario-based analysis to estimate carbon neutrality. This research provides evidence base and in-depth discussion that are supported by maps and diagrams. This research is timely and valuable. The author has conducted extensive research supported by relevant analysis methods. I have enjoyed reading this paper very much and have the following comments on this manuscript.

  1. A simple table with short descriptions of all the scenarios adopted in this research in the sub-section on scenarios under the methodology section would help.
  2. The scenarios could be named, for example, Scenario 1, Scenario 2 ….. to provide further clarity for the readers.
  3. The land areas that have been allocated for agriculture and reforestation in emission calculations for different scenarios would be useful to include in Table 1 or in a separate table.  
  4. It would be good to include some more information on how the annual costs or dollar values of carbon neutrality have been calculated. 

I look forward to seeing this article published. I wish all the very best wishes to the author for this research.

Round 2

Reviewer 1 Report

comments addressed